# Intelligent classification of platelet aggregates by agonist type

Yuqi Zhou[1†], Atsushi Yasumoto[2†], Cheng Lei[1,3]*, Chun-Jung Huang[4], Hirofumi Kobayashi[1], Yunzhao Wu[1], Sheng Yan[1], Chia-Wei Sun[4], Yutaka Yatomi[2], Keisuke Goda[1,3,5]*

[1]Department of Chemistry, University of Tokyo, Tokyo, Japan; [2]Department of Clinical Laboratory Medicine, Graduate School of Medicine, University of Tokyo, Tokyo, Japan; [3]Institute of Technological Sciences, Wuhan University, Hubei, China; [4]Department of Photonics, National Chiao Tung University, Hsinchu, Taiwan; [5]Department of Bioengineering, University of California, Los Angeles, United States

**Abstract** Platelets are anucleate cells in blood whose principal function is to stop bleeding by forming aggregates for hemostatic reactions. In addition to their participation in physiological hemostasis, platelet aggregates are also involved in pathological thrombosis and play an important role in inflammation, atherosclerosis, and cancer metastasis. The aggregation of platelets is elicited by various agonists, but these platelet aggregates have long been considered indistinguishable and impossible to classify. Here we present an intelligent method for classifying them by agonist type. It is based on a convolutional neural network trained by high-throughput imaging flow cytometry of blood cells to identify and differentiate subtle yet appreciable morphological features of platelet aggregates activated by different types of agonists. The method is a powerful tool for studying the underlying mechanism of platelet aggregation and is expected to open a window on an entirely new class of clinical diagnostics, pharmacometrics, and therapeutics.

*For correspondence:
leicheng@whu.edu.cn (CL);
goda@chem.s.u-tokyo.ac.jp (KG)

[†]These authors contributed equally to this work

## Introduction

Platelets are non-nucleated cells in blood whose principal function is to stop bleeding by forming aggregates for hemostatic reactions (*Michelson, 2012*; *George, 2000*; *Michelson, 2003*; *Harrison, 2005*). In addition to their participation in physiological hemostasis (*Michelson, 2012*; *George, 2000*; *Michelson, 2003*; *Harrison, 2005*), platelet aggregates are also involved in pathological thrombosis (*Davì and Patrono, 2007*; *Ruggeri, 2002*). Moreover, it is known that a range of diseases or medical conditions, such as inflammation, atherosclerosis, and cancer metastasis, are closely associated with platelet aggregation (*Lievens and von Hundelshausen, 2011*; *Engelmann and Massberg, 2013*; *Franco et al., 2015*; *Gay and Felding-Habermann, 2011*). Also, in patients with artificial lungs due to severe respiratory failure such as the coronavirus disease 2019 (COVID-19) caused by severe acute respiratory syndrome coronavirus 2 (SARS-CoV-2) (*Ramanathan et al., 2020*; *Ronco et al., 2020*), the long-term foreign body contact of blood with the artificial devices in the extracorporeal circulation often leads to platelet aggregation and thrombus formation followed by serious complications (e.g., myocardial infarction, cerebral infarction) (*Brodie et al., 2019*; *Brodie and Bacchetta, 2011*; *Oliver, 2009*). Here, the aggregation of platelets is elicited by a variety of agonists, which bind to and activate specific receptors expressed on the platelet. This leads to platelet activation and structural and functional changes of glycoprotein IIb/IIIa expressed on the platelet surface. The activated form of the glycoprotein can bind with fibrinogen, enabling platelets to interact with each other and form aggregates (*Michelson, 2012*; *George, 2000*; *Michelson, 2003*; *Harrison, 2005*; *Moser et al., 2008*). Despite the existence of diverse agonist types, platelet aggregates look morphologically similar and have long been thought

**eLife digest** Platelets are small cells in the blood that primarily help stop bleeding after an injury by sticking together with other blood cells to form a clot that seals the broken blood vessel. Blood clots, however, can sometimes cause harm. For example, if a clot blocks the blood flow to the heart or the brain, it can result in a heart attack or stroke, respectively. Blood clots have also been linked to harmful inflammation and the spread of cancer, and there are now preliminary reports of remarkably high rates of clotting in COVID-19 patients in intensive care units.

A variety of chemicals can cause platelets to stick together. It has long been assumed that it would be impossible to tell apart the clots formed by different chemicals (which are also known as agonists). This is largely because these aggregates all look very similar under a microscope, making it incredibly time consuming for someone to look at enough microscopy images to reliably identify the subtle differences between them. However, finding a way to distinguish the different types of platelet aggregates could lead to better ways to diagnose or treat blood vessel-clogging diseases.

To make this possible, Zhou, Yasumoto et al. have developed a method called the "intelligent platelet aggregate classifier" or iPAC for short. First, numerous clot-causing chemicals were added to separate samples of platelets taken from healthy human blood. The method then involved using high-throughput techniques to take thousands of images of these samples. Then, a sophisticated computer algorithm called a deep learning model analyzed the resulting image dataset and "learned" to distinguish the chemical causes of the platelet aggregates based on subtle differences in their shapes. Finally, Zhou, Yasumoto et al. verified iPAC method's accuracy using a new set of human platelet samples.

The iPAC method may help scientists studying the steps that lead to clot formation. It may also help clinicians distinguish which clot-causing chemical led to a patient's heart attack or stroke. This could help them choose whether aspirin or another anti-platelet drug would be the best treatment. But first more studies are needed to confirm whether this method is a useful tool for drug selection or diagnosis.

indistinguishable since the discovery of platelet aggregates in the 19th century (*Michelson, 2012*; *George, 2000*; *Michelson, 2003*; *Harrison, 2005*). This is because morphological characteristics of platelet aggregates on a large statistical scale have been overlooked as microscopy (a high-content, but low-throughput tool) has been the only method to examine them (*Finsterbusch et al., 2018*; *Nitta et al., 2018*).

In this Short Report, we present an intelligent method for classifying platelet aggregates by agonist type. This is enabled by performing high-throughput imaging flow cytometry of numerous blood cells, training a convolutional neural network (CNN) with the image data, and using the CNN to identify and differentiate subtle yet appreciable morphological features of platelet aggregates activated by different types of agonists. Our finding that platelet aggregates can be classified by agonist type through their morphology is unprecedented as it has never been reported previously. The information about the driving factors behind the formation of platelet aggregates is expected to lead to a better understanding of the underlying mechanism of platelet aggregation and open a window on an entirely new class of clinical diagnostics, pharmacometrics, and therapeutics.

## Results

### Development of the iPAC

Our procedure for developing an intelligent platelet aggregate classifier (iPAC) is schematically shown in *Figure 1A*. First, a blood sample obtained from a healthy person was separated into several different portions, into which different types of agonists were added to activate platelets while no agonist was added to the last portion for negative control (*Figure 1—figure supplement 1*; Materials and methods). Here, adenosine diphosphate (ADP), collagen, thrombin receptor activator peptide-6 (TRAP-6), and U46619 were used since they are commonly used in platelet aggregation tests (*Michelson, 2012*; *George, 2000*; *Michelson, 2003*; *Harrison, 2005*). Initially, the concentrations of the agonists were carefully chosen (20 μM for ADP, 10 μg/mL for collagen, 13 μM for TRAP-

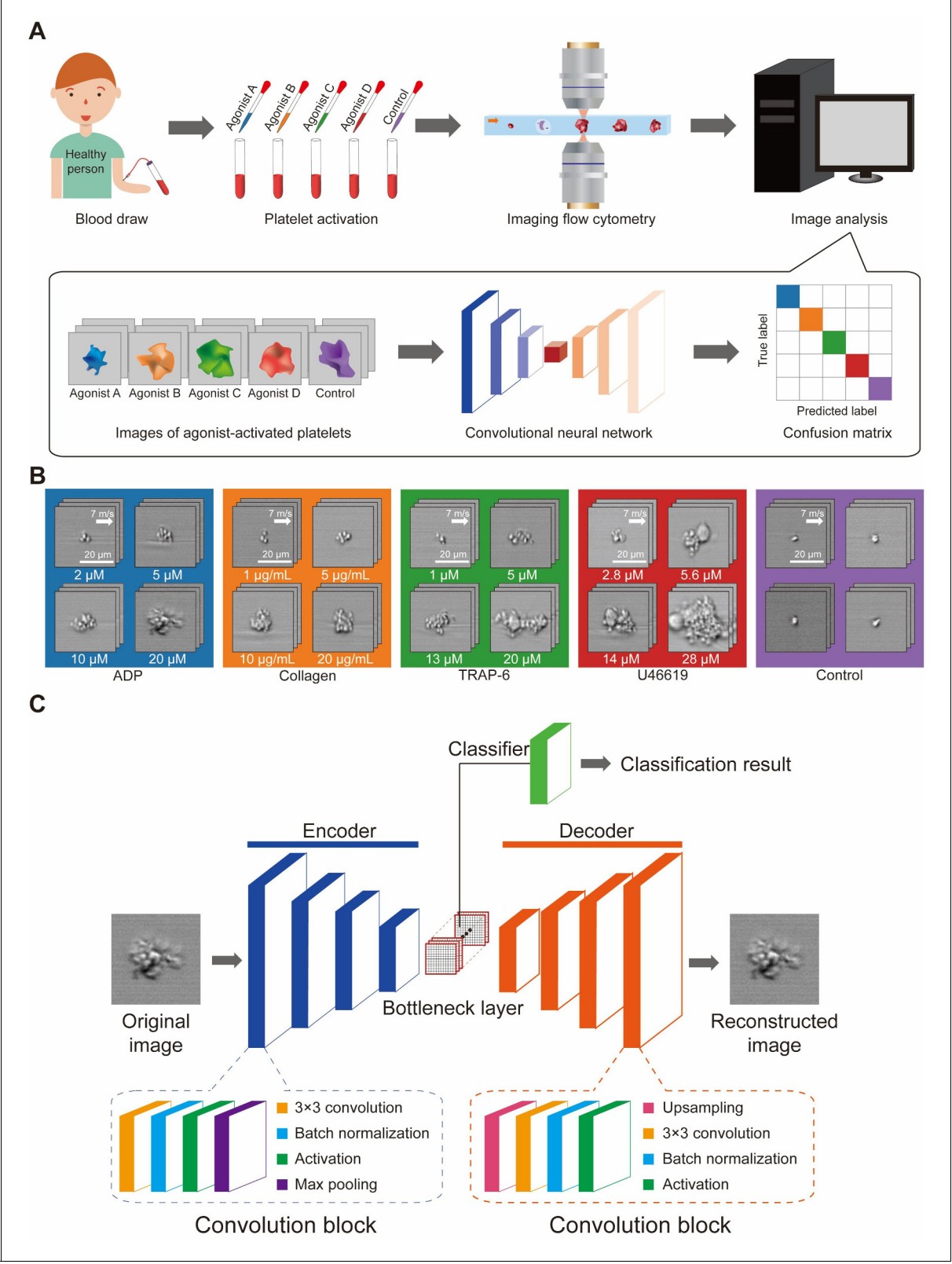

**Figure 1.** Development of the iPAC. (**A**) Procedure for developing the iPAC. (**B**) Images of the agonist-activated platelet aggregates and single platelets (negative control). (**C**) Structure of the CNN with an encoder-decoder architecture used for the development of the iPAC.
The online version of this article includes the following figure supplement(s) for figure 1:

*Figure 1 continued on next page*

**Figure supplement 1.** Detailed Procedure for Sample Preparation.
**Figure supplement 2.** Extend Image Library from Six Experimental Trials.
**Figure supplement 3.** Comparison of Different Procedures for Sample Preparation.
**Figure supplement 4.** Fabrication of the Microfluidic Chip.
**Figure supplement 5.** Schematic of the Optofluidic Time-stretch Microscope.

6, 14 µM for U46619) to minimize variations in aggregate size between the different blood sample portions. These images were acquired through six experimental trials (*Figure 1—figure supplement 2*) to mitigate potential bias in the dataset that may have come from experimental variations (e.g., signal-to-noise ratio, fluctuations in optical alignment, hydrodynamic cell focusing conditions, sample preparation). Then, four different concentrations of each agonist (2, 5, 10, 20 µM for ADP, 1, 5, 10, 20 µg/mL for collagen, 1, 5, 13, 20 µM for TRAP-6, 2.8, 5.6, 14, 28 µM for U46619) were used for platelet activation to examine the potential influence of agonist concentrations on the ability to differentiate platelet aggregates by agonist type, where the concentrations were chosen by referring to the concentrations of agonists used in light transmission aggregometry and in vitro flow-cytometric platelet aggregation tests (*Koltai et al., 2017*; *Granja et al., 2015*). The platelet aggregates were enriched by density-gradient centrifugation to remove erythrocytes from the blood sample portions. To prevent the platelet aggregates from dissolving during imaging flow cytometry, 2% paraformaldehyde was added to the blood sample portions to fix them. In addition to this sample preparation procedure, we tested other procedures such as pipetting, vortexing, fixation, and non-fixation and identified the current procedure to be advantageous over the others in preserving the morphology of platelet aggregates (*Figure 1—figure supplement 3*; Materials and methods). Second, an optofluidic time-stretch microscope (*Goda et al., 2009*; *Jiang et al., 2017*; *Lei et al., 2018*; *Lau et al., 2016*) was employed for high-throughput, blur-free, bright-field image acquisition of events (e.g., single platelets, platelet-platelet aggregates, platelet-leukocyte aggregates, single leukocytes, cell debris, remaining erythrocytes) in each sample portion (*Figure 1—figure supplements 4* and *5*; Materials and methods). Here, fluorescence image acquisition is not needed because fluorescence images of platelet aggregates would look very similar to their bright-field images (except for the colors). Third, the acquired images of the events were used to train two CNN models that classified the platelets based on their morphological features by agonist type (*Figure 1B*). Specifically, we first trained a CNN model with images of platelet aggregates activated by certain concentrations of agonists (12,000 images per agonist type) in order to examine their morphological changes while minimizing a potential influence of concentration-dependent factors on the morphology of the platelet aggregates. Then, we trained the other CNN model with a dataset in which the images of platelet aggregates activated by different concentrations of the agonists were equally mixed (12,000 images in total per agonist type) in order to show that different concentrations of the agonists do not perturb the CNN model's ability to classify platelet aggregates. We employed the CNN (*Krizhevsky et al., 2012*) with an encoder-decoder architecture to disregard insignificant features such as background noise and keep important features in the bottleneck layer and trained it with the data of a single blood donor to ensure that only the morphological features driven by the agonists contributed to the development of the iPAC (*Figure 1C*; Materials and methods). In comparison, we measured the platelet samples that were prepared under the same procedure using a conventional flow cytometer (Cytomics FC500, Beckman Coulter) which is based on fluorescence measurements for cell classification. As shown in *Figure 2*, the flow cytometer was not capable of differentiating them as indicated by their significant overlap (*Figure 2—source data 1*; Materials and methods).

## Demonstration of the iPAC

The iPAC is manifested as a confusion matrix with each row representing the examples in a predicted class and each column representing the examples in an actual or true class. As shown in *Figure 3A*, most of the images were classified into the correct groups in the diagonal line of the confusion matrix. Large separations between the different platelet sample portions in *Figure 3B* that visualizes the bottleneck layer in the CNN indicate the first CNN model's ability to discriminate various

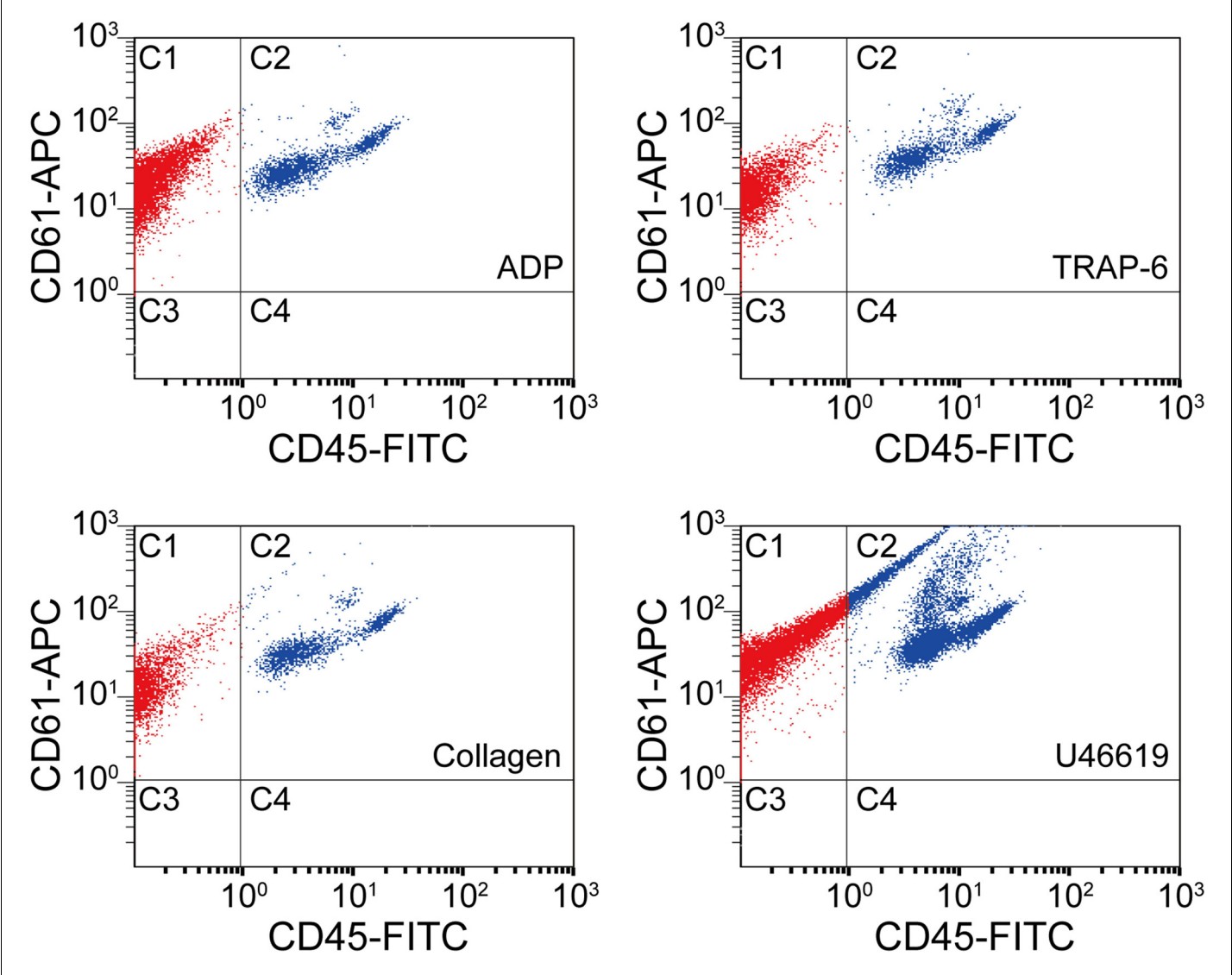

**Figure 2.** Scatter plots of agonist-activated platelets analyzed by a conventional flow cytometer. The points in region C1 are colored in red, while the points in region C2 are colored in blue for distinguishing them visually. C1: single platelets and platelet-platelet aggregates. C2: leucocytes and platelet-leucocyte aggregates. C3: blood cells other than platelets and leucocytes. C4: leucocytes.

The online version of this article includes the following source data for figure 2:

**Source data 1.** Statistical analysis of agonist-activated platelets by conventional flow cytometry.

types of agonist-activated platelet aggregates and negative control (*Figure 3—source data 1*). The negative control shows the highest classification accuracy, indicating that large morphological changes were made to the activated platelets. The U46619-treated blood sample portion shows the second highest classification accuracy of all the blood sample portions, indicating that the morphological changes caused by the agonist are very different from those caused by the other agonists. Many platelet-leukocyte aggregates were identified in the U46619-treated sample portion, but few in the other blood sample portions (*Figure 2*). This may be because U46619 acted as a thromboxane A2 (TXA$_2$) receptor agonist, which activated TXA$_2$ receptors that are abundantly expressed on platelets, vascular smooth muscle cells, and injured vascular endothelial cells. The activation of TXA$_2$ receptors may affect the morphology of U46619-induced platelet aggregates by promoting the expression of adhesion molecules that favors the adhesion and infiltration of leukocytes (*Michelson, 2012*; *George, 2000*). The low classification accuracy values of the ADP-, collagen-, and TRAP-

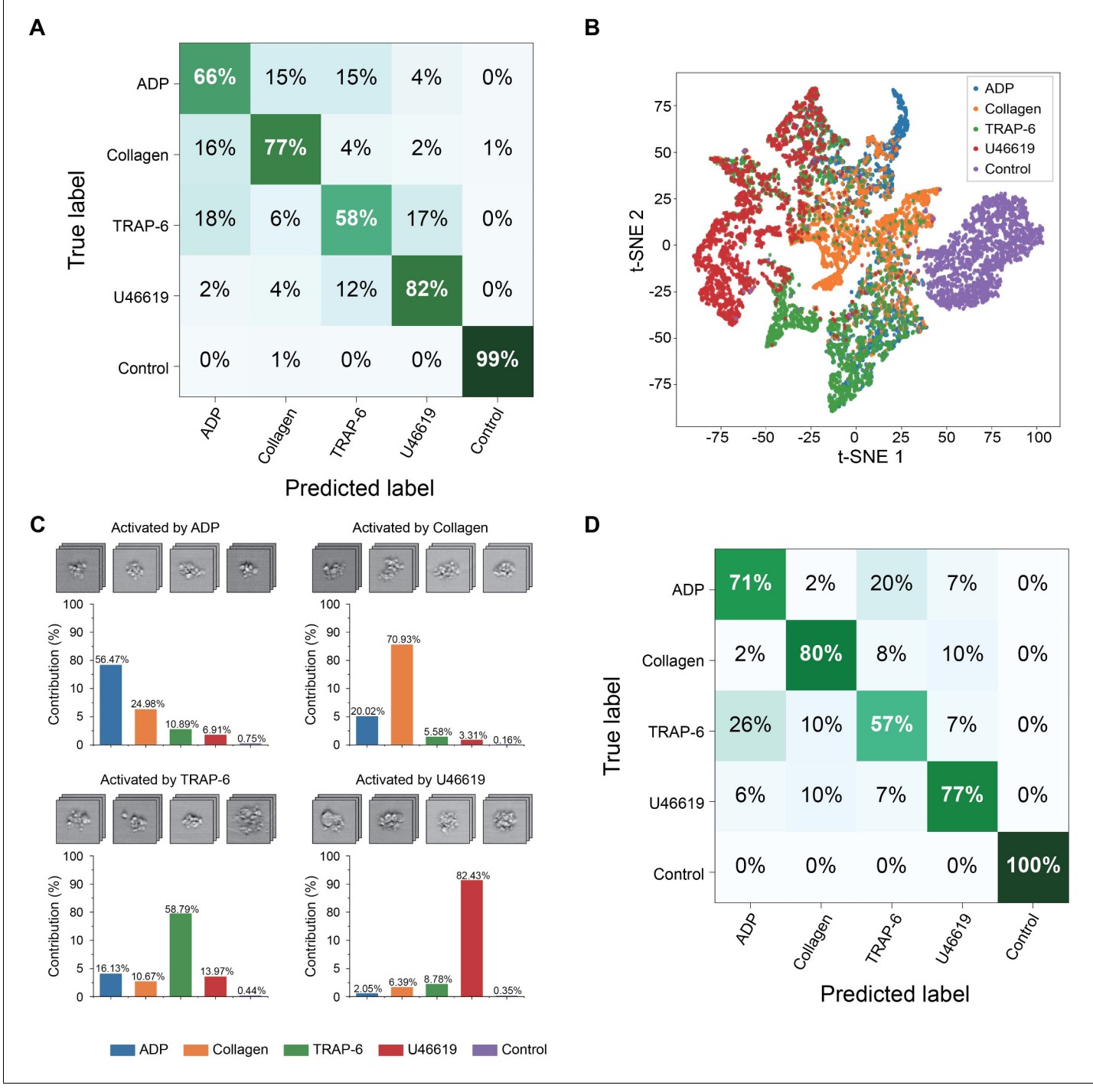

**Figure 3.** Demonstration of the iPAC. (**A**) Confusion matrix as a manifestation of the iPAC. (**B**) t-SNE plot of the agonist-activated platelet aggregates and single platelets (negative control). (**C**) Validation of the reproducibility of the iPAC. (**D**) Confusion matrix of the CNN model trained with the images of platelet aggregates activated by different concentrations of agonists.

The online version of this article includes the following source data for figure 3:

**Source data 1.** Source data of *Figure 3B*.

6-treated blood sample portions are presumably due to the fact that these agonists partially share similar mechanisms in forming platelets aggregates (*Michelson, 2012*; *George, 2000*; *Michelson, 2003*; *Harrison, 2005*; *Li et al., 2000*). For example, since platelets also release ADP

themselves during activation (*Michelson, 2012*; *George, 2000*; *Michelson, 2003*; *Harrison, 2005*), platelet aggregates produced by other agonists may also share similar morphological features as ADP-activated platelet aggregates. In addition, TRAP-6 activates thrombin receptors while thrombin generation may be amplified by other agonists during platelet activation (*Mann, 2011*), which indicates that the low prediction values of TRAP-6 can be attributed to the participation of thrombin in platelet aggregation induced by all types of agonists. Furthermore, it is common that platelets are simultaneously activated by multiple agonists whose effects on platelet aggregation are coupled whereas they are also influenced by other factors such as locally produced inhibitors, vascular endothelial cells, blood flow, and coagulation proteins during activation (*Cattaneo and Lecchi, 2007*; *Michelson, 2012*), thereby leading to the low classification accuracy values of certain agonists, which can be overcome by including the influences into the classification model to cover a wide spectrum of aggregation factors. To demonstrate the reproducibility of the iPAC, we tested it with an independent dataset (a total of 25,000 images of all event types), which was performed under the same conditions as shown in *Figure 1A*. The contribution values over all the agonists are in good agreement with the values in the diagonal elements of the confusion matrix (*Figure 3C*), which validates the reliability of the iPAC.

The iPAC's ability to classify platelet aggregates by agonist type in a concentration-independent manner is indicated by the confusion matrix shown in *Figure 3D* with an average diagonal element value of 77%. The results also reveal the existence of the unique morphological features related to each agonist type, which is promising for potential application to diagnosis of thrombotic disorders by tracing back to the leading factors of platelet aggregation. In addition, from a viewpoint of potential clinical applications, while the conventional assays can only evaluate platelet aggregability qualitatively, the iPAC can quantify it with the resolving power to identify the contribution of each agonist type to it. However, it can be recognized from the image library (*Figure 1B*) that U46619-activated platelet aggregates have relatively larger size than those in the other sample portions, which may be captured as a type of morphological features by the CNN, leading to the high classification accuracy of the U46619-activated samples.

To demonstrate the diagnostic utility of the iPAC, we applied it to blood samples of four healthy human subjects to predict the contribution of each agonist type to platelet aggregates (if any) in the samples (*Figure 4*). The blood samples were prepared by following the same procedure as shown in *Figure 1A* except for the step of adding agonists (with 2000 images of events in each blood sample). The experiment was repeated under the same conditions three times. Over 85% of the total population of platelets in all the samples were identified as single platelets, which indicates the ability of the iPAC to differentiate single platelets and platelet aggregates. Furthermore, the agonist types of the platelet aggregates in each subject's platelet classification results are consistent between the repeated experiments, indicating that the variations between the subjects resulted from platelet heterogeneity, not test variations. The iPAC's diagnostic ability to obtain this type of information is an effective tool for studying and elucidating the mechanism of platelet aggregation and holds promise for clinical diagnostics, pharmacometrics, and therapeutics, although the iPAC needs more training with a wide spectrum of diseases and medical conditions for the purpose. For example, the iPAC may provide an important clue to the choice of drugs (e.g., aspirin or thienopyridines) for antiplatelet therapy (*Mauri et al., 2014*; *Roe et al., 2012*), the gold standard of the treatment and prevention of atherothrombosis (e.g., myocardial infarction, cerebral infarction), in that aspirin inhibits the formation of $TXA_2$ whose stable analogue is U46619 while thienopyridines exert an antiplatelet effect by blocking the ADP receptor $P2Y_{12}$. Furthermore, the iPAC may be able to identify TRAP-6-activated platelet aggregates in the bloodstream of patients with deep vein thrombosis (since TRAP-6 interacts with the receptor of thrombin) and suggest that they come from the venous side.

## Discussion

The information about the driving factors behind the formation of platelet aggregates is expected to lead to a better understanding of the underlying mechanisms of platelet aggregation and, thereby, open a window on an entirely new class of clinical diagnostics and therapeutics. For example, antiplatelet therapy is the gold standard of the treatment and prevention of atherothrombosis (e.g., myocardial infarction, cerebral infarction) for which aspirin and thienopyridines (e.g., prasugrel

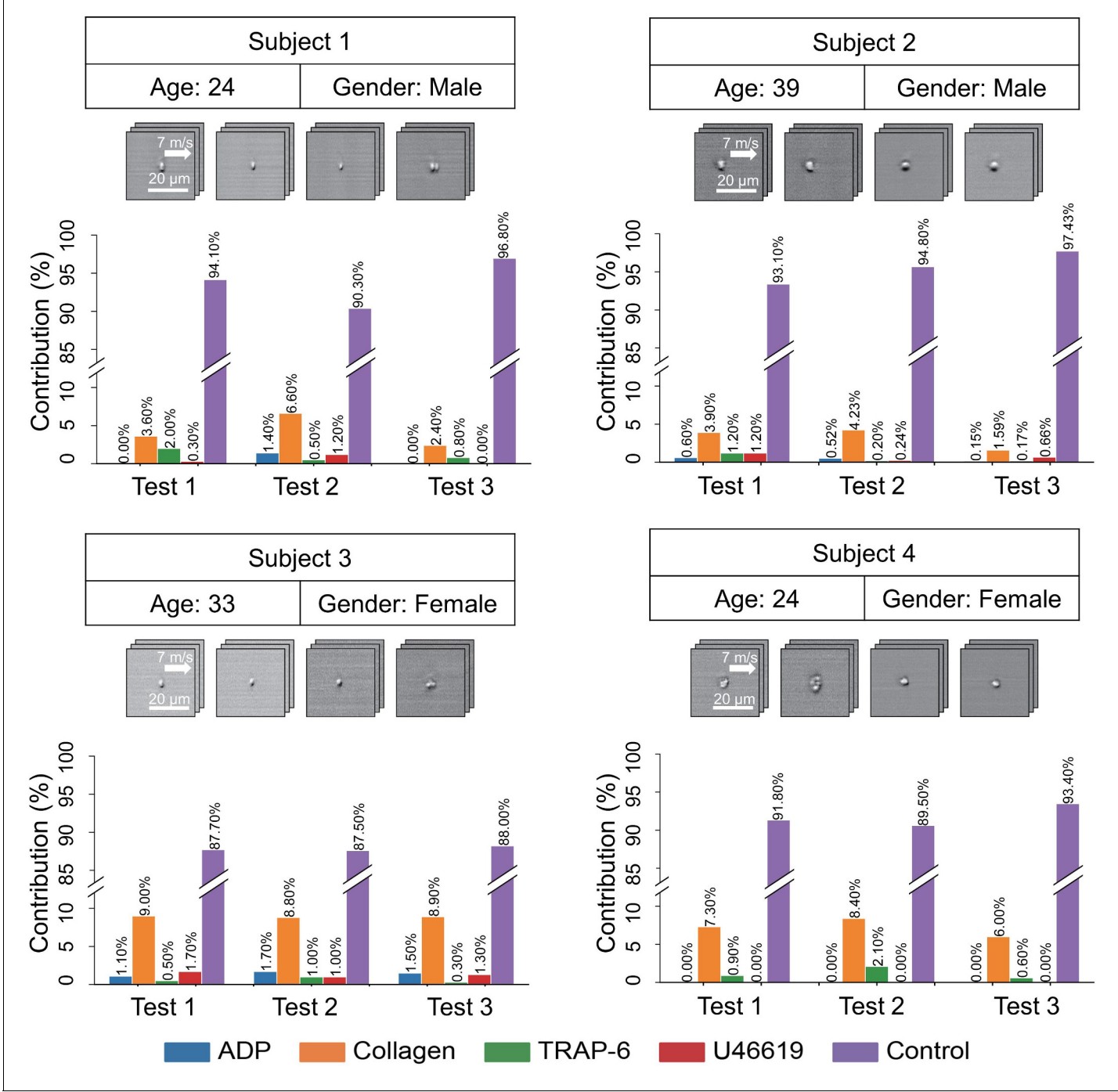

**Figure 4.** iPAC-based diagnosis of platelets from four healthy human subjects. The experiment was repeated under the same conditions three times per subject.

and clopidogrel) are primarily used as antiplatelet drugs worldwide (*Mauri et al., 2014*; *Roe et al., 2012*). Aspirin inhibits the formation of $TXA_2$ whose stable analogue is U46619, whereas thienopyridines exert an antiplatelet effect by blocking the ADP receptor $P2Y_{12}$. Accordingly, the ability to identify the type of platelet aggregates in the blood stream may provide an important clue to the choice of a drug for antiplatelet therapy. Furthermore, deep vein thrombosis (DVT) is a blood clot that normally occurs in a deep vein where coagulation activation plays an important role. Since TRAP-6 interacts with the receptor of thrombin (i.e., the product of the coagulation cascade), the

ability to identify TRAP-6-activated platelet aggregates in the blood stream may suggest that aggregates come from the venous side. Therefore, the iPAC may pave the way for introducing a novel laboratory testing technique for the management of pathological thrombosis such as atherothrombosis and DVT although further basic and clinical studies are needed.

The relation between platelet activation signaling pathways and the formation of platelet aggregates has been extensively studied (*Li et al., 2010*; *Michelson, 2012*; *Brass et al., 2013*). It is known that agonists activate platelets in a selective manner via specific receptors, which is followed by a variety of downstream signaling events (*Li et al., 2010*). For example, collagen interacts with the immune-like receptor glycoprotein VI, which signals through an immunoreceptor tyrosine-based activation motif and activates the tyrosine phosphorylation pathway (*Michelson, 2012*; *Li et al., 2010*) In contrast, soluble agonists such as TRAP-6, U46619, and ADP interact with G protein-coupled receptors (*Michelson, 2012*; *Brass, 2003*). Furthermore, each soluble agonist couples with a specific type of G protein, which leads to different aggregation mechanisms (*Rivera et al., 2009*) and thus suggests different underlying mechanisms for expressing different morphological features on platelet aggregates. It is challenging, but is expected to be intriguing to study and elucidate the mechanisms for a further understanding of the biology of platelets.

# Materials and methods

## Key resources table

| Reagent type (species) or resource | Designation | Source or reference | Identifiers | Additional information |
|---|---|---|---|---|
| Antibody | PE anti-human CD61 (mouse monoclonal) | BioLegend | Cat#336405; RRID:AB_1227583 | Platelet samples (5 µL per sample) |
| Antibody | Conjugated Antibody CD45-FITC | Beckman Coulter | Cat#A07782; RRID:AB_10645157 | FACS (5 µL per test) |
| Antibody | APC Mouse Anti-Human CD61 (mouse monoclonal) | BD Pharmingen | Cat#564174; RRID:AB_2738645 | FACS (5 µL per test) |
| Chemical compound, drug | Collagen | HYPHEN BioMed | Cat#AG005K-CS | Platelet activation |
| Chemical compound, drug | Revohem ADP | Sysmex | Cat#AP-200–422 | Platelet activation |
| Chemical compound, drug | TRAP-6 amide trifluoroacetate salt | BACHEM | Cat#H-2936.0005 | Platelet activation |
| Chemical compound, drug | U46619 | Cayman Chemical | Cat#16450 | Platelet activation |
| Chemical compound, drug | 4% Paraformaldehyde Phosphate Buffer Solution | WAKO | Cat#30525-89-4 | Fixation (2% Paraformaldehyde) |
| Chemical compound, drug | Lymphoprep | STEMCELLS | Cat#ST07851 | Density-gradient medium |
| Chemical compound, drug | KMPR 1035 | MicroChem | Cat#Y211066 | Negative photoresist |
| Chemical compound, drug | SU-8 Developer | MicroChem | Cat#Y020100 | Developer |
| Chemical compound, drug | SYLGARD 184 Silicone Elastomer | Dow Corning | Cat#1064291 | Microfluidic device |

*Continued on next page*

*Continued*

| Reagent type (species) or resource | Designation | Source or reference | Identifiers | Additional information |
|---|---|---|---|---|
| Software, algorithm | Matlab | MathWorks | https://mathworks.com/products/matlab.html | Image recovery |
| Software, algorithm | Keras | others | https://github.com/keras-team/keras | Python library; Image analysis |
| Software, algorithm | Tensorflow | arXiv:1603.04467 | https://arxiv.org/abs/1603.04467 | Python library; Image analysis |
| Software, algorithm | AutoCAD | Autodesk | https://www.autodesk.com/products/autocad/overview | Microfluidic channel design |

## Blood samples for detection of platelet aggregates

The detailed procedure of the sample preparation is shown in *Figure 1—figure supplement 1*, where platelets and platelet aggregates were enriched from whole blood by the density-gradient centrifugation to maximize the detection efficiency (*Beakke, 1951*). Specifically, blood samples were obtained from a healthy person with 3.2% citric acid as the anticoagulant (*Figure 1—figure supplement 1A*). Although it has a depressed concentration of ionized calcium, 3.2% citrate blood is desirable for clinical coagulation tests (*Adcock et al., 1997*; *Cazenave et al., 2004*). The other common anticoagulants, such as heparin and ethylenediaminetetraacetic acid (EDTA), are not suitable for this study because they influence the coagulation functions of platelets (*Ludlam, 1981*). Platelets were immunofluorescently labeled by adding 20 μL PE anti-human CD61 (BioLegend, 336405) to the blood samples to ensure that platelets would be detected in all images (*Figure 1—figure supplement 1B*). For each agonist type, 500 μL blood was incubated with 50 μL agonist solution, which contained 20 μM ADP (BioMed, AP-200–422), 10 μg/mL Collagen (BioMed, AG005K-CS), 13 μM TRAP-6 (H2936.0005, BACHEM), or 14 μM U46619 (Cayman Chemical, 16450), for 10 min (*Figure 1—figure supplement 1C*). The labeled, activated platelets were then diluted using 5 mL saline (*Figure 1—figure supplement 1D*). Next, the platelets were isolated by using Lymphoprep (STEM-CELLS, ST07851), a density-gradient medium, using the protocol provided by the vendor. Specifically, the diluted blood was added on top of the Lymphoprep and then centrifuged at 800 g for 20 min (*Figure 1—figure supplement 1E*). After the centrifugation, 1 mL of the sample was taken from the mononuclear layer, to which 1 mL of 2% paraformaldehyde (Wako, 163–20145) was added for fixation (*Lanier and Warner, 1981*; *Figure 1—figure supplement 1F,G*). The operation of the fixation was performed at 4°C for 30 min while other operations were performed at 25°C room temperature. As shown in *Figure 1—figure supplement 2*, we first compared several procedures of preparing blood samples, but most of the procedures either left a large amount of non-target blood cells in the sample, thus decreasing the iPAC's detection efficiency, or dismantled the agonist-activated platelet aggregates. The current procedure is advantageous over the procedures in preserving the morphology of platelet aggregates while eliminating non-target blood cells. This study was approved by the Institutional Ethics Committee in the School of Medicine at the University of Tokyo [no. 11049-(6)]. Written informed consents were obtained from the blood donors.

## Microfluidic chip fabrication

The microfluidic chip was fabricated using standard photolithographic methods (*Whitesides et al., 2001*). A designed pattern of the microfluidic channel was drawn using AutoCAD (Autodesk) and printed on a film mask (UnnoGiken). Negative photoresist (KMPR 1035, MicroChem) was spin-coated on a silicon wafer and heated at 100°C for 10 min. Then, the silicon wafer, covered with the film mask, was exposed to ultraviolet (UV) light followed by hard baking at 100°C for 5 min and developed using SU-8 developer (MicroChem). After washing with isopropyl alcohol and water, the silicon wafer was heated at 150°C for 15 min. The negative photoresist mold on the silicon wafer was fixed in a Petri dish and then filled with polydimethylsiloxane (PDMS, Dow Corning) in which PDMS base and curing reagent were mixed at a ratio of 10:1 (*Figure 1—figure supplement 4A*). PDMS was heated at 80°C for 15 min, and then a small piece of coverslip was placed on PDMS right above the

observation area of the microfluidic channel. This step improved the mechanical strength of PDMS so that the channel (*Figure 1—figure supplement 4B*) was able to resist the pressure inside the channel without deformation. After another heating for more than 1 hr, the PDMS layer was cut into a small piece so that it could fit in the size of a glass slide (*Figure 1—figure supplement 4C*). The inlets and outlet were punched by a 25G needle (*Figure 1—figure supplement 4D*). To form permanent bonding between the PDMS channel and the glass slide, both the PDMS device and the glass slide were treated with a plasma cleaner (Harrick Plasma) (*Figure 1—figure supplement 4E*). The dimensions of the microchannel in the observation area are about 80 μm in width and 40 μm in height (*Figure 1—figure supplement 4F*).

## Optofluidic time-stretch microscopy

The optofluidic time-stretch microscope (*Lei et al., 2016*) is schematically shown in *Figure 1—figure supplement 5*. A Ti:Sapphire mode-locked femtosecond pulse laser with a center wavelength, bandwidth, and pulse repetition rate of 780 nm, 40 nm, and 75 MHz, respectively, was used as an optical source. Each laser pulse was first stretched temporally by a single-mode dispersive fiber with a group-velocity dispersion of −240 ps/nm (Nufern 630-HP) and then dispersed spatially by the first diffraction grating with a groove density of 1200 lines/mm. The stretched laser pulse was focused by the first objective lens (Olympus, 40×, NA 0.6) onto a flowing cell in the microfluidic channel. The pulse that contained the spatial profile of the cell on its spectrum was collected by the second objective lens and spatially recombined by the second diffraction grating, followed by photodetection with a high-speed photodetector (New Focus 1580-B) with a detection bandwidth of 12 GHz. To ensure imaging of platelet-related events (i.e., single platelets, platelet-platelet aggregates, platelet-leukocyte aggregates), fluorescence detection was used in conjunction with the optofluidic time-stretch microscope. A 488 nm continuous-wave laser was used to detect CD61 fluorescence signals with a photomultiplier tube (Hamamatsu H10723-01MOD). Only the image signals associated with CD61 fluorescence signals were collected. The image-encoded pulse and fluorescence signal were digitized using a high-speed oscilloscope (Tektronix DPO 71604B) with a detection bandwidth of 16 GHz and a sampling rate of 50 GS/s. Pulses were repeated by the mode-locked pulse laser at 75 MHz so that image-encoded pulses detected by the photodetector were digitally stacked to form 2D images using MATLAB R2018b (MathWorks). The pulse intensity profile (usually Gaussian-shaped with ripples) was normalized to obtain a flat background. Also, all the images were cropped into 160 × 160 pixels by the same cropping algorithm, by which the cell-contained part was completely included in each image for further analysis.

## Evaluation of agonist-activated platelets by conventional flow cytometry

We analyzed agonist-activated platelets with a conventional flow cytometer (Cytomics FC500, Beckman Coulter) that can count and analyze large cell populations via scattering and fluorescence measurements with high throughput. Blood samples were processed using the same procedure as for optofluidic time-stretch microscopy, but labeled with anti-CD61-APC and anti-CD45-FITC antibodies (Beckman Coulter) for detecting white blood cells and platelets, respectively. To only detect single platelets and platelets aggregates, gating of cellular size and granularity was applied to the light scatter plots. As shown in *Figure 2*, the C1 areas, which correspond to CD61-APC positive and CD45-FITC negative, show events associated with single platelets and platelet-platelet aggregates. The C2 areas, which correspond to CD61-APC/CD45-FITC double positive, show events associated with platelet-leukocyte aggregates. The C3 areas (CD61-APC/CD45-FITC double negative) and C4 areas (CD61-APC negative and CD45-FITC positive) correspond to events which did not contain any platelets.

## Convolutional neural network

The details of the CNN with the encoder-decoder architecture are as follows. The encoder was used to extract morphological features of platelet aggregates, while the decoder was used to recover the platelet aggregate images from the morphological features. This two-stage structure forced the encoder to extract features from the cells instead of the background or noise, which helped enhance the reliability and accuracy of classification. The images were normalized to 0-mean and divided into

training, validation, and test sets at a ratio of 3:1:1. The CNN classifier was trained on the training set. The validation loss was calculated with the validation dataset at each epoch to monitor the learning process. The learning rate was reduced when the validation loss stopped descending for more than 3 epochs until it reached $1 \times 10^{-8}$. The training was ceased when there was no more decrease in the validation loss for more than 6 epochs. After the training ended, the test set was processed to calculate the final classification accuracy for each agonist type. The CNN classifier was implemented on Keras (*Chollet, 2015*) with the Tensorflow (*Abadi et al., 2016*) backbone. The training of the CNN classifier was optimized by Adam with an initial learning rate of 0.001.

## Acknowledgements

This work was supported by the ImPACT Program, JSPS Core-to-Core Program, White Rock Foundation, Nakatani Foundation, and University of Tokyo's Center for Nano Lithography.

## Additional information

### Competing interests

Keisuke Goda: is a shareholder of two cell analysis startups (CYBO and Cupido). The other authors declare that no competing interests exist.

### Funding

| Funder | Grant reference number | Author |
|---|---|---|
| Government of Japan | ImPACT Program | Yuqi Zhou<br>Atsushi Yasumoto<br>Cheng Lei<br>Chun-Jung Huang<br>Hirofumi Kobayashi<br>Yunzhao Wu<br>Sheng Yan<br>Chia-Wei Sun<br>Yutaka Yatomi<br>Keisuke Goda |
| Japan Society for the Promotion of Science | Core-to-Core Program | Yuqi Zhou<br>Atsushi Yasumoto<br>Cheng Lei<br>Chun-Jung Huang<br>Hirofumi Kobayashi<br>Yunzhao Wu<br>Sheng Yan<br>Chia-Wei Sun<br>Yutaka Yatomi<br>Keisuke Goda |
| White Rock Foundation | | Yuqi Zhou<br>Atsushi Yasumoto<br>Cheng Lei<br>Chun-Jung Huang<br>Hirofumi Kobayashi<br>Yunzhao Wu<br>Sheng Yan<br>Chia-Wei Sun<br>Yutaka Yatomi<br>Keisuke Goda |
| Nakatani Foundation | | Yuqi Zhou<br>Atsushi Yasumoto<br>Cheng Lei<br>Chun-Jung Huang<br>Hirofumi Kobayashi<br>Yunzhao Wu<br>Sheng Yan<br>Chia-Wei Sun<br>Yutaka Yatomi<br>Keisuke Goda |

| University of Tokyo | Center for Nano Lithography | Yuqi Zhou<br>Atsushi Yasumoto<br>Cheng Lei<br>Chun-Jung Huang<br>Hirofumi Kobayashi<br>Yunzhao Wu<br>Sheng Yan<br>Chia-Wei Sun<br>Yutaka Yatomi<br>Keisuke Goda |
|---|---|---|

The funders had no role in study design, data collection and interpretation, or the decision to submit the work for publication.

## Author contributions

Yuqi Zhou, Data curation, Formal analysis, Validation, Investigation; Atsushi Yasumoto, Conceptualization, Data curation, Validation; Cheng Lei, Conceptualization, Resources, Supervision, Methodology; Chun-Jung Huang, Formal analysis, Methodology; Hirofumi Kobayashi, Data curation, Formal analysis; Yunzhao Wu, Data curation, Formal analysis, Validation; Sheng Yan, Data curation; Chia-Wei Sun, Resources, Supervision; Yutaka Yatomi, Conceptualization, Resources, Supervision; Keisuke Goda, Conceptualization, Resources, Supervision, Funding acquisition, Project administration

## Author ORCIDs

Yuqi Zhou https://orcid.org/0000-0002-1206-0049
Cheng Lei https://orcid.org/0000-0001-8439-4235
Hirofumi Kobayashi http://orcid.org/0000-0002-4505-2061
Keisuke Goda https://orcid.org/0000-0001-6302-6038

## Ethics

Human subjects: This study was approved by the Institutional Ethics Committee in the School of Medicine at the University of Tokyo [no. 11049-(6)]. Written informed consents were obtained from the blood donors.

## Decision letter and Author response

Decision letter https://doi.org/10.7554/eLife.52938.sa1
Author response https://doi.org/10.7554/eLife.52938.sa2

# Additional files

## Supplementary files

• Transparent reporting form

## Data availability

Image data has been deposited in Dryad (https://doi.org/10.5061/dryad.fn2z34tpz).

The following dataset was generated:

| Author(s) | Year | Dataset title | Dataset URL | Database and Identifier |
|---|---|---|---|---|
| Zhou Y, Yasumoto A, Lei C, Huang C-J, Kobayashi H, Wu Y, Yan S, Sun C-W, Yatomi Y, Goda K | 2019 | Intelligent classification of platelet aggregates by agonist type | http://doi.org/10.5061/dryad.fn2z34tpz | Dryad Digital Repository, 10.5061/dryad.fn2z34tpz |

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
