## [Decision Letter]

**Acceptance summary:**

The manuscript describes a methodology to analyze platelet (and leukocyte) aggregates that utilizes microfluidics, optofluidic time-stretch microscopy and convolutional neural network analysis. The manuscript presents data that the technique can distinguish activation by four platelets agonist – ADP, collagen, U46619 and TRAP-6.

**Decision letter after peer review:**

Thank you for submitting your article "Intelligent classification of platelet aggregates by agonist type" for consideration by *eLife*. Your article has been reviewed by Aleksandra Walczak as the Senior Editor, a Reviewing Editor, and two reviewers. The reviewers have opted to remain anonymous.

The reviewers have discussed the reviews with one another and the Reviewing Editor has drafted this decision to help you prepare a revised submission.

Summary:

The authors present a novel deep learning based method to classify platelet aggregation based on various agonists using high-throughput imaging. The sample preparation and the advantages of the procedure is well explained. The capture images are used to train and eventually classify images of platelet aggregates.The manuscript presents data that the technique can distinguish activation by four platelets agonist – ADP, collagen, U46619 and TRAP-6.

A number of questions have been raised. We invite you to submit a revised version addressing these.

Essential revisions:

1) Data are presented for three healthy subjects. One donor has a different pattern than the rest. The authors interpret this as showing that the test could have diagnostic utility. But all three donors where healthy, thus I interpret this as variation in the test. There is no other assessment of the variation amongst normals.

2) Endothlelial TXA2 receptors are cited (subsection “Demonstration of the iPAC”) as a mechanism of one effect from U46619, but they studied studied blood, which likely has, if anything, a very small amount of circulating endothelial cells.

3) It is unlikely that a platelet will ever see a single agonist in circulation. This is not discussed. Collagen-induced activation of platelets results in ADP and TXA2 release. They tested all three of those agonists. Some consideration of these combinations would aid in the feasibility of this novel technology. Similarly, platelets circulate in an environment with locally produced inhibitors (NO, prostacyclins, ADPases). Their effect has not been assessed, and this can be substantial (see Cattaneo et al., 2007).

4) Platelets circulate in vivo where the calcium concentration is 2 mM. The studies are done in citrated plasma where the calcium concentration is far lower. Studying platelets at artificially low calcium concentrations has, in the past, led to artifactual findings. Other anticoagulants could be used.

5) Was there additional activation of the platelets when they were spun on the gradient?

6) In deep learning network used for classification what is the need for the up-sampling layer (decoder part)? The decode part reconstructs the image. Is this reconstructed image being used anywhere? Why is the classifier layer at the CNN bottleneck not sufficient for classification?

7) The authors mention that crops from the actual image containing the cells are used for classification. However, how these crops are generated is a big question. Are they manually cropped from the image or some automated technique is used to obtain the cropped regions, specifically during testing phase?

8) The classification accuracy for the TRAP-6 in both confusion matrix of Figure 3 is low in comparison to the others. Is there any explanation on why so? Is there any class imbalance during training phase? How many total images were used for training and how mane images were there in each class?

9) Accuracy of which data is shown in Figure 3A and 3C?

10) A more elaborate explanation on what can be seen in the images in Figure 1—figure supplement 3 will be good. Also, can the images be shown at same pixel size/scale?

11) Has the classification accuracy been compared with any other multi-class classification method from literature?

12) As I understand, the classifier is trained using brightfield images. If the fluorescence images being used to capture any extra information for classification?

[Editors' note: further revisions were suggested prior to acceptance, as described below.]

Thank you for resubmitting your work entitled "Intelligent classification of platelet aggregates by agonist type" for further consideration by *eLife*. Your revised article has been evaluated by Aleksandra Walczak (Senior Editor) and a Reviewing Editor.

The manuscript has been improved but there are some remaining issues that need to be addressed before acceptance, as outlined below:

1) Comment 2: Please change "endothelial TXA2 receptors" to "TXA2 receptors" to avoid confusion.

2) In response to comment 3, the authors did not address this concern about the lack of locally produced inhibitors (NO, PGI2, ectoADPases etc): "Similarly platelets circulate in an environment with locally produced inhibitors (NO, prostacyclins, ADPases). Their effect has not been assessed, and this can be substantial (see Cattaneo et al., 2007)."

3) In response to comment 4, the authors appear to misstate Cazenave et al., 2004 which says "Citrate is the preferred anticoagulant for blood collection,.…. however, this method has certain disadvantages. In particular, the PRP preparation has a limited stability (no longer than 2 hours) and contains plasma proteins, including enzymes. In addition, human platelet-rich plasma (PRP) prepared from blood collected into trisodium citrate (3.8% w/v) has a depressed ionic calcium concentration, which can cause platelet aggregation and release of substances during centrifugation (2). To overcome these different problems, a centrifugation technique has been developed for the isolation and washing of platelets from human or rodent blood anticoagulated with acid-citrate-dextrose (ACD). The cells are resuspended in a physiological buffer under well-defined conditions, notably the presence of plasmatic ionic calcium concentrations (2 mM) and the absence of coagulation factors or other plasma components". Thus, physiological calcium (2 mM) needs to be added back to the platelet suspension to avoid artifacts. If the authors wish to study platelet-rich plasma, an alternate non-calcium chelating anti-coagulant, such as PPACK could be used.

Please make textual changes to your manuscript to indicate these caveats to your data.

---

## [Author Response]

Essential revisions:1) Data are presented for three healthy subjects. One donor has a different pattern than the rest. The authors interpret this as showing that the test could have diagnostic utility. But all three donors where healthy, thus I interpret this as variation in the test. There is no other assessment of the variation amongst normals.

We thank the reviewer for the comment. We interpreted the data from the three healthy subjects as individual variations, not test variations, because all three donors are healthy, but their genders, ages, and races are different. To provide more solid evidence, we have performed additional experiments to obtain results from additional healthy subjects and repeat the experiments on different days three times. As shown in revised Figure 3E (now Figure 4 in the revised manuscript), each subject’s platelet classification results are consistent, indicating that the variations between the subjects result from the individual variations, not test variations. To address the reviewer’s comment, we have revised/added the following text in the revised manuscript (subsection “Demonstration of the iPAC”): “To demonstrate the diagnostic utility of the iPAC, we applied it to blood samples of four healthy human subjects to predict the contribution of each agonist type to platelet aggregates (if any) in the samples (Figure 4). The blood samples were prepared by following the same procedure as shown in Figure 1A except for the step of adding agonists (with 2,000 images of events in each blood sample). The experiment was repeated under the same conditions three times. Over 85% of the total population of platelets in all the samples were identified as single platelets, which indicates the ability of the iPAC to differentiate single platelets and platelet aggregates. Furthermore, the agonist types of the platelet aggregates in each subject’s platelet classification results are consistent between the repeated experiments, indicating that the variations between the subjects resulted from platelet heterogeneity, not test variations.”

2) Endothlelial TXA2 receptors are cited (subsection “Demonstration of the iPAC”) as a mechanism of one effect from U46619, but they studied studied blood, which likely has, if anything, a very small amount of circulating endothelial cells.

We thank the reviewer for the comment. As they correctly state, the concentration of circulating endothelial cells in blood is very small. However, TXA2 receptors on platelet membranes and vascular endothelial cells are abundantly expressed, such that they may have a significant effect on U46619-induced platelet aggregation. To clarify this point, we have added the following statement to the revised manuscript (subsection “Demonstration of the iPAC”): “U46619 acted as a thromboxane A2 (TXA2) receptor agonist to activate endothelial TXA2 receptors that are abundantly expressed on platelets, vascular smooth muscle cells, and injured vascular endothelial cells. The activation of TXA2 receptors may affect the morphology of U46619-induced platelet aggregates by promoting the expression of adhesion molecules that favors the adhesion and infiltration of leukocytes (Michelson, 2012; George, 2000).”

3) It is unlikely that a platelet will ever see a single agonist in circulation. This is not discussed. Collagen-induced activation of platelets results in ADP and TXA2 release. They tested all three of those agonists. Some consideration of these combinations would aid in the feasibility of this novel technology. Similarly, platelets circulate in an environment with locally produced inhibitors (NO, prostacyclins, ADPases). Their effect has not been assessed, and this can be substantial (see Cattaneo et al., 2007).

We thank the reviewer for the comment. It is true that a platelet in vivo is activated by multiple agonists simultaneously. This is why the diagonal values in the confusion matrices shown in Figure 3A and Figure 3D are not 100%, but there exist some cross-coupling effects from multiple agonists on platelet aggregation. Our ultimate goal is to develop a complete classification model that can predict the contribution of each agonist very accurately, even when multiple agonists influence platelet aggregation simultaneously. This requires image data from experiments in which multiple agonists are used to induce platelet aggregation both in vitro and in vivo. To clarify this point, we have added the following statement to the revised manuscript (subsection “Demonstration of the iPAC”): “Furthermore, it is common that platelets are simultaneously activated by multiple agonists whose effects on platelet aggregation are coupled, thereby leading to the low classification accuracy values of certain agonists, which can be overcome by including the influences of multiple agonists into the classification model to cover a wide spectrum of aggregation factors.”

4) Platelets circulate in vivo where the calcium concentration is 2 mM. The studies are done in citrated plasma where the calcium concentration is far lower. Studying platelets at artificially low calcium concentrations has, in the past, led to artifactual findings. Other anticoagulants could be used.

We thank the reviewer for the comment. In our study, we used citric acid because it is a reversible calcium chelator, does not significantly reduce the calcium concentration, and is commonly used in the study of platelets and coagulation proteins. Alternatively, ACD can be used because it is an anticoagulant that contains citric acid and is equivalent to citrated blood. On the other hand, other anticoagulants such as heparin and EDTA are not suitable for this study because heparin attenuates platelet function by the inhibition of thrombin while EDTA is a strong irreversible calcium chelator that completely suppresses platelet function. To clarify this point, we have added the following statement to the revised manuscript (Discussion section): “specifically, blood samples were obtained from a healthy person with citric acid as the anticoagulant, which did not dissociate platelet aggregates (Figure 1—figure supplement 1A) while maintaining the sufficient calcium concentration for platelet aggregation (Cazenave, 2004). The other common anticoagulants, such as heparin and ethylenediaminetetraacetic acid (EDTA), are not suitable for this study because they influence the coagulation functions of platelets (Ludlam, 1981).”

5) Was there additional activation of the platelets when they were spun on the gradient?

We thank the reviewer for the comment. The high percentage (99%) of the “control” in the confusion matrix indicates that platelets were not significantly affected by the sample preparation procedure. Furthermore, even if there is a minor influence from the density gradient on the morphology of platelet aggregates, it should be considered identical among all the samples since we applied the same protocol to each sample.

6) In deep learning network used for classification what is the need for the up-sampling layer (decoder part)? The decode part reconstructs the image. Is this reconstructed image being used anywhere? Why is the classifier layer at the CNN bottleneck not sufficient for classification?

We thank the reviewer for the comment. The reconstructed images were used in the training process to optimize the CNN’s performance. By minimizing the difference between the original images and the reconstructed images, we ensured in the CNN model that the features in the bottleneck layer were crucial to cellular morphology. This two-stage structure forced the encoder to extract features from the cells instead of the background or noise (while the classification result may be prone to noise if only the classifier is applied in the CNN), which helped us enhance the reliability and accuracy of platelet classification.

7) The authors mention that crops from the actual image containing the cells are used for classification. However, how these crops are generated is a big question. Are they manually cropped from the image or some automated technique is used to obtain the cropped regions, specifically during testing phase?

We thank the reviewer for the comment. The images were cropped in the image recovery process automatically. In the process, the region of the cell in each image is identified by using a segmentation algorithm. Then, the image was cropped into 160 x 160 pixels in size with the cell region at the center of the image. Therefore, the cell region remains intact in the image. This cropping algorithm was the same for all the images in this work. To clarify this point, we have added the following statement to the revised manuscript (subsection “Blood samples for detection of platelet aggregates”): “Also, all the images were cropped into 160×160 pixels by the same cropping algorithm, by which the cell-contained part was completely included in each image for further analysis.”

8) The classification accuracy for the TRAP-6 in both confusion matrix of Figure 3 is low in comparison to the others. Is there any explanation on why so? Is there any class imbalance during training phase? How many total images were used for training and how mane images were there in each class?

We thank the reviewer for the comment. The low classification accuracy elements of the TRAP-6-treated blood sample portions in the confusion matrix are presumably due to the fact that ADP, collagen, and TRAP-6 agonists partially share similar mechanisms in forming platelets aggregates, as discussed in previous publications (Michelson, 2012; George, 2000; Michelson, 2003; Harrison, 2005; Li et al., 2000). In particular, TRAP-6 activates protease-activated receptor-1 (PAR-1), which is one of the thrombin receptors. Meanwhile, thrombin is released into the blood when platelets are activated by the other agonists. Therefore, TRAP-6-induced platelet aggregates may be similar to those induced by other agonists due to the thrombin-related platelet aggregation. As for the number of images for training, the classes are balanced during the training process. In fact, 12,000 images for each class were used for the training (60,000 images in total). To clarify the point of the low TRAP-6 element value, we have added the following statement to the revised manuscript (subsection “Demonstration of the iPAC”): “In addition, TRAP-6 activates thrombin receptors while thrombin generation may be amplified by other agonists during platelet activation (Mann, 2011), which indicates that the low prediction values of TRAP-6 can be attributed to the participation of thrombin in platelet aggregation induced by all types of agonists. Furthermore, it is common that platelets are simultaneously activated by multiple agonists whose effects on platelet aggregation are coupled, thereby leading to the low classification accuracy values of certain agonists, which can be overcome by including the influences of multiple agonists into the classification model to cover a wide spectrum of aggregation factors.”

9) Accuracy of which data is shown in Figure 3A and 3C?

We thank the reviewer for the comment. Figure 3A shows the accuracy of the images of platelet aggregates induced by the optimized concentrations of agonists (20 µM for ADP, 10 µg/mL for collagen, 13 µM for TRAP-6, 14 µM for U46619), while Figure 3C shows the images and prediction results of an independent dataset that was not used in the training process. The independent dataset was acquired from the blood samples induced by the optimized concentrations of agonists (20 µM for ADP, 10 µg/mL for collagen, 13 µM for TRAP-6, 14 µM for U46619) under the same preparation procedure.

10) A more elaborate explanation on what can be seen in the images in Figure 1—figure supplement 3 will be good. Also, can the images be shown at same pixel size/scale?

We thank the reviewer for the suggestion. We have added more text to the legend of the supplementary figure and revised the figure with images at the same pixel size and scale.

11) Has the classification accuracy been compared with any other multi-class classification method from literature?

We thank the reviewer for the comment. To the best of our knowledge, this is the first time that we classify platelet aggregates by morphology and there is no previous work that shows classification of platelet aggregates.

12) As I understand, the classifier is trained using brightfield images. If the fluorescence images being used to capture any extra information for classification?

We thank the reviewer for the comment. Although we have not done high-throughput fluorescence imaging of platelets, we doubt that the fluorescence images will provide extra information for classification because fluorescence does not differentiate the contribution of each agonist and fluorescence images of platelet aggregates will look pretty much similar to their bright-field images (except for the colors). Since different types of leukocytes can be stained with different fluorescent probes, platelet-leukocytes can be better classified with the availability of their fluorescent images, but fluorescent labeling is not desirable for this work because it also activates platelets and may produce artifacts. To clarify this point, we have added the following statement to the revised manuscript (subsection “Development of the iPAC”): “Here, fluorescence image acquisition is not needed because fluorescence images of platelet aggregates would look very similar to their bright-field images (except for the colors).”

[Editors' note: further revisions were suggested prior to acceptance, as described below.]

The manuscript has been improved but there are some remaining issues that need to be addressed before acceptance, as outlined below:1) Comment 2: Please change "endothelial TXA2 receptors" to "TXA2 receptors" to avoid confusion.

We thank the reviewers for the comment. We have made the change as requested.

2) In response to comment 3, the authors did not address this concern about the lack of locally produced inhibitors (NO, PGI2, ectoADPases etc): "Similarly platelets circulate in an environment with locally produced inhibitors (NO, prostacyclins, ADPases). Their effect has not been assessed, and this can be substantial (see Cattaneo et al., 2007)."

We thank the reviewers for the comment. It is true that the locally produced inhibitors have a substantial effect on platelet aggregation. However, to evaluate this effect using our iPAC, an experiment has to be conducted in vivo, where locally produced inhibitors, vascular endothelial cells, blood flow, and coagulation proteins affect platelet aggregation simultaneously. In this manuscript, the purpose of our study is to classify platelet aggregates by agonist based on our intelligent classification model that was constructed by adding various agonists into whole blood in vitro, measuring their effects on the morphology of platelet-platelet aggregates and platelet-leukocyte aggregates, and forming the confusion matrix as a linear superposition of the effects. Thus, the low classification accuracy values of some agonists in the confusion matrices (Figure 3A, Figure 3D) are also attributed to the fact that in vitro experiments are not able to evaluate all factors of platelet aggregation. In other words, it is meaningful to perform this study in vitro because if these agonists are administrated in vivo to develop the confusion matrix, it can lead to thrombosis and death. On the other hand, we hope to perform in vivo experiments using animals and address this point in our future study. To clarify this point, we have revised the following statement in the manuscript (subsection “Demonstration of the iPAC”): “Furthermore, it is common that platelets are simultaneously activated by multiple agonists whose effects on platelet aggregation are coupled whereas they are also influenced by other factors such as locally produced inhibitors, vascular endothelial cells, blood flow, and coagulation proteins during activation (Cattaneo, 2007; Michelson, 2012), thereby leading to the low classification accuracy values of certain agonists, which can be overcome by including the influences into the classification model to cover a wide spectrum of aggregation factors.”

3) In response to comment 4, the authors appear to misstate Cazenave et al., 2004 which says "Citrate is the preferred anticoagulant for blood collection,.…. however, this method has certain disadvantages. In particular, the PRP preparation has a limited stability (no longer than 2 h) and contains plasma proteins, including enzymes. In addition, human platelet-rich plasma (PRP) prepared from blood collected into trisodium citrate (3.8% w/v) has a depressed ionic calcium concentration, which can cause platelet aggregation and release of substances during centrifugation (2). To overcome these different problems, a centrifugation technique has been developed for the isolation and washing of platelets from human or rodent blood anticoagulated with acid-citrate-dextrose (ACD). The cells are resuspended in a physiological buffer under well-defined conditions, notably the presence of plasmatic ionic calcium concentrations (2 mM) and the absence of coagulation factors or other plasma components". Thus, physiological calcium (2 mM) needs to be added back to the platelet suspension to avoid artifacts. If the authors wish to study platelet-rich plasma, an alternate non-calcium chelating anti-coagulant, such as PPACK could be used.

We thank the reviewers for the comment. Considering the practical utility of the iPAC, we used the same conditions as those used in conventional clinical coagulation tests in Japan, where 3.2% citrate blood is used. In addition, as described in the original manuscript, platelet aggregation was induced by adding various agonists into whole blood, instead of PRP used by Cazenave et al., before the centrifugation was performed. The use of washed platelets in PRP eliminates the involvement of leukocytes and coagulation factors and is significantly different from the mechanism of platelet aggregation in vivo, and therefore, we think it is important to do experiments with whole blood. Although the calcium concentration in citrate blood may not be the same with that in vivo, we expect the iPAC constructed based on in vitro experiments to be practically useful as a new laboratory testing tool for diagnosing thrombosis, and therefore, 3.2% citrate whole blood (not an alternate noncalcium chelating anticoagulant) was used in our experiments as it can minimize the problem of a depressed ionic calcium concentration. Meanwhile, we also consider the effect of the anticoagulant and ionic calcium concentration on the morphology of platelet aggregates as an exciting topic to address in the future. To clarify these points, we have revised the manuscript as follows (subsection “Blood samples for detection of platelet aggregates”): “Specifically, blood samples were obtained from a healthy person with 3.2% citric acid as the anticoagulant (Figure 1—figure supplement 1A). Although it has a depressed concentration of ionized calcium, 3.2% citrate blood is desirable for clinical coagulation tests (Adcock, 1997; Cazenave, 2004).”